# Self-FI: Self-Supervised Learning for Disease Diagnosis in Fundus Images

**DOI:** 10.3390/bioengineering10091089

**Published:** 2023-09-16

**Authors:** Toan Duc Nguyen, Duc-Tai Le, Junghyun Bum, Seongho Kim, Su Jeong Song, Hyunseung Choo

**Affiliations:** 1Department of AI Systems Engineering, Sungkyunkwan University, Suwon 16419, Republic of Korea; austin47@g.skku.edu; 2College of Computing and Informatics, Sungkyunkwan University, Suwon 16419, Republic of Korea; 3Sungkyun AI Research Institute, Sungkyunkwan University, Suwon 16419, Republic of Korea; 4Department of Ophthalmology, Kangbuk Samsung Hospital, School of Medicine, Sungkyunkwan University, Suwon 16419, Republic of Korea; 5Biomedical Institute for Convergence, Sungkyunkwan University, Suwon 16419, Republic of Korea; 6Department of Electrical and Computer Engineering, Sungkyunkwan University, Suwon 16419, Republic of Korea

**Keywords:** self-supervised learning, medical image processing, fundus images

## Abstract

Self-supervised learning has been successful in computer vision, and its application to medical imaging has shown great promise. This study proposes a novel self-supervised learning method for medical image classification, specifically targeting ultra-wide-field fundus images (UFI). The proposed method utilizes contrastive learning to pre-train a deep learning model and then fine-tune it with a small set of labeled images. This approach reduces the reliance on labeled data, which is often limited and costly to obtain, and has the potential to improve disease detection in UFI. This method employs two contrastive learning techniques, namely bi-lateral contrastive learning and multi-modality pre-training, to form positive pairs using the data correlation. Bi-lateral learning fuses multiple views of the same patient’s images, and multi-modality pre-training leverages the complementary information between UFI and conventional fundus images (CFI) to form positive pairs. The results show that the proposed contrastive learning method achieves state-of-the-art performance with an area under the receiver operating characteristic curve (AUC) score of 86.96, outperforming other approaches. The findings suggest that self-supervised learning is a promising direction for medical image analysis, with potential applications in various clinical settings.

## 1. Introduction

Deep learning has established itself as a cornerstone technique in computer vision, driving significant advancements in various applications, including but not limited to object detection, segmentation, and classification [1]. Despite its powerful capabilities, these deep learning models are not without their challenges. They necessitate a substantial quantity of labeled data to deliver a high degree of accuracy [2]. The acquisition of such data, particularly in the domain of medical imaging, can pose a substantial challenge, both in terms of feasibility and associated costs [3]. One potential solution to mitigate this dependency on labeled data is self-supervised learning (SSL). This approach harnesses the inherent structure and context intrinsic to the data to learn meaningful representations [4]. Conceptually, SSL falls under the broader category of unsupervised learning, wherein the model is trained to predict certain aspects of the data based on its own learned representations, thereby effectively reducing the requirement for external annotations [5,6]. SSL methods have already showcased compelling results in the field of natural image processing [7]. Consequently, the prospect of their successful application within medical imaging is a compelling avenue that merits further exploration [8].

Ultra-wide-field fundus images (UFI) and conventional fundus images (CFI) are pivotal diagnostic instruments in the field of ophthalmology, providing indispensable insights into the retina, thereby facilitating the diagnosis and subsequent management of an array of ocular conditions [9,10,11]. UFI images afford an expansive glimpse of the retina, capturing up to 200 degrees, and thus offering a more comprehensive field of view compared to CFI images, which are limited to a range of 30 to 50 degrees of the retina [12,13]. Despite their advantages, diagnosing ocular diseases from UFI images presents several complexities. These challenges stem from the intrinsic intricacies of the images themselves, the requirement of expert personnel for accurate interpretation, and the scarcity of suitably labeled data for machine learning purposes [14]. The evolution of accurate and efficient methodologies for disease diagnosis from UFI images holds immense potential. With such advances, patient outcomes could be markedly improved, aiding in earlier diagnosis, more precise management plans, and, consequently, ultimately enhancing overall prognosis.

Image classification in medical imaging involves the automatic assignment of predefined labels to medical images based on their visual content. In the context of this study, the term “image classification” is used to refer to the automated categorization of medical images, which serves a role analogous to preliminary disease diagnosis. Existing literature highlights the potential of self-supervised learning (SSL) in medical imaging, specifically its role in enhancing the accuracy and minimizing dependence on labeled data. Despite this, the majority of self-supervised learning methods have predominantly focused on natural images, which encompass everyday scenes and objects often used in computer vision research. This has resulted in a research gap pertaining to the application of SSL techniques to the domain of ultra-wide-field fundus images (UFI) in the context of medical imaging [15]. This paper, therefore, embarks on an exploration of SSL performance in the domain of medical image classification using UFI images [16]. Among various SSL approaches, contrastive learning has garnered considerable attention as an efficacious strategy to derive useful representations from unlabeled data [7]. Its core principle rests on maximizing the similarity between various augmented perspectives of an identical image while concurrently reducing the similarity amongst augmented views of disparate images. Accomplishing this necessitates the design and implementation of several loss functions such as InfoNCE loss and NT-Xent loss [17,18]. Recent studies have expanded upon this by investigating the effectiveness of employing various augmentation strategies—including but not limited to cutout, mixup, and random erasing—to enhance the performance of contrastive learning [19]. Furthermore, certain investigations have adopted a task-specific approach, tailoring contrastive learning to particular tasks such as object detection and image segmentation, and yielding promising results in the process [20,21,22]. Overall, these pioneering studies not only validated the effectiveness of contrastive learning as a powerful tool for unsupervised learning but also lay a robust foundation for prospective research in this captivating field.

In this paper, we introduce an SSL strategy that harnesses contrastive learning principles to train a deep learning model for disease classification in ultra-wide-field fundus images (UFI). The proposed methodology, self-FI, employs bi-lateral contrastive learning and multi-modality pre-training to extract the correlation between UFI and conventional fundus images (CFI), illustrated in Figure 1. We pre-train the model on an extensive array of unlabeled UFI and CFI images before fine-tuning it on a smaller, labeled UFI dataset. To evaluate the effectiveness of the proposed method, we utilize a publicly accessible dataset and compare our approach against other prevalent techniques. Our pivotal contributions in this work are multifaceted and are outlined as follows:We demonstrate that our self-supervised learning method outperforms conventional supervised learning in UFI classification, achieving unrivaled results with an AUC score of 86.96. This underlines the potential of self-supervised learning as an effective instrument for medical image analysis, especially in contexts where labeled data are either scarce or prohibitively expensive to procure.We propose a novel contrastive learning technique that employs a new loss function, thereby enhancing the quality of the learned representations. Our approach draws upon both intra- and inter-image correlations to derive a feature space that is optimally suited for UFI classification. We substantiate the effectiveness of our method through a series of experiments and validations, including an examination of learned feature maps and an assessment of different hyperparameters.We offer comprehensive heatmaps of our models for each of the six diseases encompassed in our dataset using GradCAM [23]. This provides an in-depth insight into the model’s focus on each disease. We validate our model by juxtaposing it against a range of other methodologies, including supervised learning and transfer learning from preexisting models. Our proposed model outperforms these comparative methods, emphasizing the crucial role of self-supervised learning in medical image analysis.Our work signifies a significant leap forward in the application of self-supervised learning for disease diagnosis in retinal fundus images. By harnessing the capabilities of contrastive learning and our innovative loss function, we demonstrate the feasibility of achieving state-of-the-art results with a limited quantity of labeled data. Our approach presents a novel pathway for medical practitioners to diagnose patients efficiently and accurately, thereby enhancing patient outcomes and the overall quality of care.

The remainder of this paper is organized as follows: Section 2 provides detail about our dataset and novel methodology, leveraging bi-lateral contrastive learning and multi-modality pre-training. Section 3 presents the results, showcasing our model’s performance compared to other techniques, and includes an analysis of our model’s disease focus via heatmaps and evaluation. Finally, Section 4 encapsulates our discussion and conclusions, reflecting on our findings, the implications of our proposed method, and potential directions for future research in this promising field.

## 2. Materials and Methods

### 2.1. Dataset

The dataset used in this study, which was collected from KangBuk Samsung Hospital, comprised ultra-wide-field fundus images (UFI) and conventional fundus images (CFI) from 2 January 2006 to 31 December 2019. To respect patient privacy and confidentiality, all personally identifiable information was removed from the images before use. This study adhered to the tenets of the Declaration of Helsinki, and the protocol was reviewed and approved by the Institutional Review Boards (IRB) of Kangbuk Samsung Hospital (No. KBSMC 2020-01-031-001). This is a retrospective study of medical records, and our data were fully anonymized. Therefore, the IRB waived the requirement for informed consent. Based on the correlations, we formed sub-datasets from the dataset, shown in Figure 2, as follows:UFI unlabeled: The UFI unlabeled subset is a diverse collection of 15,706 ultra-wide-field fundus images, including UFI images from UFI Left–Right and UFI–CFI Pair. Without any specific annotations, these images span across a variety of eye conditions and also include images of healthy eyes. This collection is an invaluable asset for self-supervised and unsupervised learning approaches.UFI Left–Right: Comprising 8382 fundus images of various eye conditions and healthy eyes, the UFI Left–Right pair dataset is specifically assembled to assist tasks that require data in pairs, such as stereo image matching. Each image in this set has a corresponding left and right pair.UFI–CFI: The UFI–CFI pair subset consists of 7166 fundus images, incorporating both ultra-wide-field fundus images (UFI) and conventional fundus images (CFI). Created to explore the benefits of multi-modality pre-training through self-supervised learning, this set spans across all diseases and healthy eye conditions, thereby serving as a rich resource for disease diagnostic research.UFI Labeled: The UFI labeled subset is a curated collection of ultra-wide-field fundus images, annotated by a team of experienced ophthalmologists, and consists a total of 3285 images. Primarily intended for supervised learning approaches that require labeled data for training, these annotations offer vital details about the presence and severity of various eye diseases, including diabetic retinopathy, age-related macular degeneration, and glaucoma.

### 2.2. Methodology

In this study, we use a two-step methodology to improve the classification performance in retinal disease diagnosis, as shown in Figure 3. The first step is to perform self-supervised pre-training, where the objective is to identify and learn features inherent in medical fundus images. For this, we employ a contrastive learning approach using three unique types of image pairs: pair-instance, bi-lateral, and multi-modality. This step allows the model to capture the general features and structures within fundus images, but without focusing on disease-specific characteristics. Once the model is proficient in recognizing these features, we transition to the second phase, which is supervised fine-tuning. During this phase, the pre-trained model is further refined on a labeled dataset specifically designed for retinal disease diagnosis. This fine-tuning ensures that the model, initialized with features learned from fundus images, becomes adept at identifying and classifying retinal diseases. In essence, the first phase prepares the groundwork through broad feature learning, while the second phase tailors the model for the specific task of disease diagnosis.

Contrastive learning is a self-supervised approach that leverages the inherent structure of data to learn meaningful representations without relying on explicit labels. Central to this paradigm are the concepts of positive and negative pairs. A positive pair typically consists of two augmented versions of the same data instance, implying that they share a common underlying source of information and, therefore, should be close in the learned feature space. Conversely, a negative pair comprises two distinct data instances, suggesting that they come from different sources and should thus be far apart in the feature space. The objective of self-contrastive learning is to pull positive pairs close together while pushing negative pairs apart in a representation space, as shown in Figure 2. To this end, the InfoNCE loss, or noise-contrastive estimation loss, is often employed. InfoNCE loss quantifies the similarity between representations of positive pairs in relation to a set of negative samples. By minimizing this loss, models are trained to recognize and emphasize the differences between positive and negative pairs, ultimately leading to a rich and discriminative feature space that captures the essential characteristics of the data.

In general, the generation of a positive pair is initiated by selecting an individual image *u* from the dataset. Subsequently, two distinct data augmentations are applied to this image to produce two correlated but unique views $u_{i}$ and $u_{j}$. These augmented views serve as a positive pair and undergo further processing through a neural network architecture. Initially, both $u_{i}$ and $u_{j}$ are passed through an encoder *f*, which in our case is implemented using a ResNet architecture. The encoder transforms the input images into high-dimensional feature vectors $f(u_{i})$ and $f(u_{j})$. To facilitate the subsequent contrastive learning, a projection head *g* is employed to map these feature vectors into a representation space where contrastive loss is more stable and easier to optimize. Mathematically, the image representation *z* is obtained as $z_{i}=g\left(f\left(u_{i}\right)\right)$. The projection head *g* itself is a neural network layer characterized by its weight matrix *W*. It receives the encoder’s output $f(u_{i})$ and maps it through a ReLU non-linearity ω to produce the final output representation *h*, given by $h=\omega (W\cdot f\left(z_{i}\right))$. Figure 4 illustrates the framework of pair-instance, bi-lateral, and multi-modality, where bi-lateral and multi-modality not only use one individual *u*, but rather in two pairs ($u_{l}-u_{r}$ and u−c) to form positive samples.

#### 2.2.1. Pair-Instance Pre-Training

For pair-instance pre-training, we employ data augmentation methods, similar to SimCLR [18], to create a positive pair. This method focuses on distinguishing between pairs of differently augmented instances derived from the same image. The underlying goal of SimCLR is to amplify the similarity between representations stemming from two unique augmentations of a single image while concurrently minimizing any similarity present among the representations of distinct images. This approach presents a stark contrast to traditional supervised learning methodologies that rely heavily on explicitly provided labels, thereby allowing SimCLR to learn directly from the inherent structure of the data. SimCLR follows a systematic two-stage training regimen, with the first stage dedicated to feature extraction from the input image using a neural network encoder, specifically the ResNet34 [24]. The second stage is oriented towards the transformation of this high-dimensional feature vector into a space of lower dimensionality using a projection head. This component consists of one or more fully connected layers, each bolstered with non-linear activations.

Given an image *u*, two distinct augmentations $u_{i}$ and $u_{j}$ are applied to generate two correlated, yet distinct, views $u_{i}$ and $u_{j}$ of the same image. The aim is to maximize the similarity between the encoded projections of the positive pair, relative to all other negative pairs in the same batch. First, in the encoding stage, each image *u* from the positive pair is passed through a neural network base encoder f(·) and projection head g(·). The representation results are represented as $z_{i}$ and $z_{j}$. The similarity metric $sim(z_{i},z_{j}$) is defined using the cosine similarity as:(1)$$sim\left(z_{i},z_{j}\right)=\frac{z_{i}\cdot z_{j}}{{∥z_{i}∥}_{2}\times {∥z_{j}∥}_{2}}$$
where $z_{i}\cdot z_{j}$ represents the dot product of $z_{i}$ and $z_{j}$, and ${∥z_{i}∥}_{2}\times {∥z_{j}∥}_{2}$ represents the L2 norms (Euclidean norms). The output of the cosine similarity ranges between −1 and 1, where a value close to 1 implies that the vectors are similar, and a value close to −1 implies that the vectors are dissimilar.

The operational pipeline of SimCLR commences with encoding two distinct input images—both being post data augmentation—using an encoder base network denoted as f(·). This step results in the generation of feature representations that are further transformed by a projection head g(·). Central to SimCLR is the presumption that a pair of images created by applying various augmentations to a single source image share a common ground of information and, hence, are considered a positive pair. However, this approach does not come without limitations. The acquisition of knowledge remains confined to a single image, thus restricting the model’s ability to harness wider contextual understanding. To address this constraint, we propose two novel strategies for contrastive pre-training: bi-lateral pre-training and multi-modality pre-training. The former entails training with left–right eye images from ultra-wide-field fundus (UFI) images, thereby acknowledging the inherent symmetry in anatomical structures and the potential commonalities in pathological occurrences. In contrast, the latter exploits the combination of UFI and conventional fundus Images (CFI) for training, leveraging the differing perspectives and resolutions of these image types.

#### 2.2.2. Bi-Lateral Pre-Training

For fundus images, it is common to have a pair of two images from the same patient, capturing the left and right body parts of the patient. These images are presumed to encapsulate identical patient-specific information, albeit presenting differing views and image structures. This characteristic proves advantageous for the model, as it facilitates learning of representations from multiple angles and viewpoints, thus enhancing its understanding of image patterns. This is the basis of our bi-lateral pre-training, where we leverage the left and right eye images from ultra-wide-field fundus imaging, treating them as a positive pair. The nuanced differences between these paired images equip the model with a broader perspective and greater feature comprehension, which ultimately refines its performance in subsequent disease classification tasks. The inherent differences between these paired images, which span distinct angles and capture varied structural intricacies, imbue the model with a broader perspective. It exposes the model to diverse and complex feature variations, which contribute towards its ability to understand and interpret patterns more effectively. By incorporating this information from paired images into its learning process, the model is equipped with a more nuanced understanding of the data. This understanding extends beyond what could be derived from an individual image, thereby enabling the model to better distinguish between normal and pathological conditions.

Suppose $u_{l}$ and $u_{r}$ represent the left and right eye images, respectively, from ultra-wide-field fundus (UFI) imaging of a given patient. The aim of the bi-lateral pre-training is to maximize the similarity between these two representations. This can be formulated as a contrastive loss function:(2)$$L_{Bi-lateral}=-\log\frac{exp(sim\left(z_{l},z_{r}/\tau \right)}{\sum_{k=1}^{K}exp\left(sim\left(z_{l},z_{k}\right)/\tau \right)}$$
where f(·) is the encoder network that maps an image to a feature representation, sim(·) is a similarity function such as cosine similarity, *K* is the total number of negative samples, and τ is a temperature parameter that adjusts the concentration of the distribution. In this setup, the encoder network f(·) is trained to extract features such that the representations of the left and right eye images ($z_{l}$ and $z_{r}$) are similar. This is achieved by minimizing the contrastive loss $L_{Bi-lateral}$. For a given positive pair $\left(u_{l},u_{r}\right)$ in the dataset, the loss aims to increase the similarity between $z_{l}$ and $z_{r}$ while reducing the similarity between $z_{l}$ and $z_{k}$, where $z_{k}$ is a negative sample that is not related to $z_{l}$. The objective is to learn a robust representation that captures patient-specific information shared between left and right eye images while distinguishing between different patients.

Bi-lateral pre-training is designed to augment the model’s grasp of the unique attributes present within ultra-wide-field fundus images (UFI). By presenting the model with paired left–right eye images, we aim to empower it with a broader perspective that captures intricate variations within the images. This technique enriches the model’s ability to discern nuanced textures, patterns, and structural elements, which in turn contribute to a more comprehensive understanding of the dataset. Although the immediate focus is on image comprehension, the gained insights indirectly enhance the model’s capacity for accurate disease classification during subsequent fine-tuning.

#### 2.2.3. Multi-Modality Pre-Training

For the dual-modality pre-training, we introduce another contrastive learning strategy that capitalizes on the combination of UFI and CFI. Each type of imaging provides unique and complementary information: whereas UFIs offer a wider field of view encompassing peripheral regions, CFIs offer a focused, detailed visualization of the vasculature. Suppose *U* and *C* are the UFI and CFI images, respectively, of the same patient. The objective of the dual-modality pre-training is to maximize the similarity between these two representations. This can be expressed mathematically using the contrastive loss function:(3)$$L_{multi-modality}=-\log\frac{exp(sim\left(z_{U},z_{C}/\tau \right)}{\sum_{k=1}^{K}exp(sim\left(z_{U},z_{k}/\tau \right)}$$

In this formulation, f(·) is again the encoder network mapping an image to a feature representation, sim(·) is the similarity function such as cosine similarity, *K* is the total number of negative samples, and τ is the temperature parameter. In this scenario, we train the encoder network f(·) to extract features such that the representations of the UFI and CFI images ($z_{U}$ and $z_{C}$) are similar. The contrastive loss $L_{multi-modality}$ is minimized during training. The idea is to learn a robust representation that effectively fuses the complementary information from UFI and CFI images, thereby enhancing the model’s overall understanding and classification capabilities.

The rationale behind training UFI and CFI together lies in the recognition that each image modality holds distinct attributes that complement one another. UFI images encapsulate a panoramic view of the retina, revealing features that may extend beyond the scope of conventional images. Conversely, CFI images, although limited in their field of view, offer high-resolution details that might not be as prevalent in UFI images. By training the model on this diverse combination, we aim to exploit the unique advantages of each modality to provide a comprehensive understanding of retinal structures. The multi-modality approach enriches the model’s ability to capture the intricate variations and patterns that characterize retinal images, which subsequently contributes to more effective disease classification during the fine-tuning phase.

Dual-modality pre-training fundamentally relies on the premise that the UFI and CFI images, despite their differing scopes, contain overlapping and mutually informative data about a patient’s condition. By maximizing the similarity between the representations of UFI and CFI images, the model can take advantage of these shared insights, ultimately enhancing the versatility and robustness of the learned representations. Moreover, this approach benefits from the distinct attributes of UFI and CFI images. The UFI’s wide-field view is particularly useful in diseases where peripheral retinal information is crucial. On the other hand, CFI’s high-resolution imaging of the central retina is beneficial for diseases predominantly affecting the central field. By coupling these two modalities, the model has access to a more holistic understanding of the patient’s condition, broadening its diagnostic capabilities. The effectiveness of dual-modality pre-training is reliant on the careful selection of the similarity function, sim(·), and the optimization of the temperature parameter, τ. These choices are pivotal in maintaining the balance between the model’s ability to distinguish the informative patterns in the data and its capacity to avoid overfitting. Furthermore, the selection of negative samples in the denominator of the contrastive loss function is a key factor influencing the model’s performance. In the context of dual-modality pre-training, negative samples should ideally be chosen to encourage the model to distinguish between different patients, rather than merely different imaging modalities.

#### 2.2.4. Self-FI: Self-Supervised Learning Model on Retinal
Fundus Images for Disease Diagnosis

Our proposed methodology harnesses the power of three distinctive pre-training approaches: pair-instance, bi-lateral, and multi-modality pre-training, as shown in Figure 5. This unique amalgamation of techniques serves to learn high-quality representations from a diverse, unlabeled dataset of UFI and CFI images. In the SimCLR pre-training scheme, a pair of augmented views derived from the same image forms the “positive” pair. Bi-lateral pre-training further enriches the model’s understanding by treating each pair of left–right UFI images as a “positive” instance, thus allowing the model to appreciate the inherent similarities in bi-lateral anatomical structures. To exploit the complementary information embedded in different imaging modalities, we design multi-modality pre-training, where a “positive” pair comprises a UFI and a CFI image from the same patient. Contrastive learning thrives not just on the identification of “positive” pairs, but also on the distinction from “negative” pairs. To this end, for every “positive” pair, we introduce a “negative” pair composed of an image randomly selected from the remainder of the dataset. This forces the model to discern between images exhibiting similar and dissimilar features. The success of our method hinges on a novel loss function that amalgamates the losses from each of the pre-training strategies. The final loss, $L_{total}$, is the sum of the losses from SimCLR, bi-lateral, and multi-modality pre-training, each weighted by a coefficient optimized through rigorous hyperparameter tuning. This composite loss function is represented as:(4)$$L_{total}=L_{pair-instance}+L_{bi-lateral}+L_{multi-modality}$$

The method is summarized in Algorithm 1.
**Algorithm 1:** Proposed contrastive learning methods.
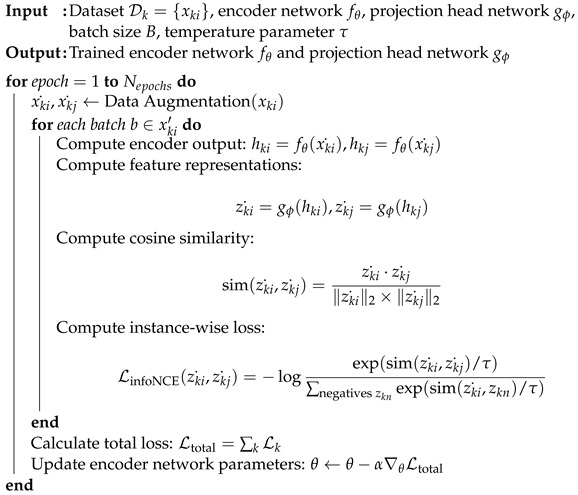


### 2.3. Implementation Settings

To assess the effectiveness of our proposed contrastive learning approaches, we deploy a ResNet34 neural network as our foundational model, denoted by f(·), paired with a projection head g(·). This projection head comprises a fully connected layer sequence: Linear → ReLu → Linear. This sequence maps the high-dimensional image feature vectors, generated by f(·), into a lower-dimensional feature space. For model training, we employ the AdamW optimizer with a learning rate of 0.00005, weight decay of 0.0001, and temperature of 0.07. The model is trained over 3000 epochs with a batch size of 128 and a 128-dimensional embedding. In the pre-training phase, images are resized to 224 × 224 pixels, and a range of augmentation techniques are used to yield positive pairs. These techniques encompass horizontal and vertical flipping, random cropping, rotation (up to 180 degrees), color adjustment (brightness, contrast, saturation, hue), grayscale application, and Gaussian blurring. Each pre-training sub-dataset is divided into 80:20 for training and validating, respectively. We utilize the SimCLR framework for pair-instance pre-training, bi-lateral pre-training, and multi-modality pre-training. The implementation of our proposed method is constructed using Pytorch [25] and is executed on four GeForce RTX 2080 Ti GPUs. These GPUs afford the necessary computational resources to manage the expansive dataset and the complexity of the model. The GPUs’ parallel processing capability enables us to train our model effectively and expedite pre-training using the proposed methods. Once the foundational model is trained, we leverage it to fine-tune the downstream multi-label classification task. We retain the pre-trained weights of the ResNet34 while substituting the projection head with a fully connected layer. This layer yields an output vector with a size corresponding to the number of classes represented in the dataset. For this phase, we train the model for 100 epochs, with a batch size of 64. The model is optimized using the AdamW optimizer with a learning rate of 0.001 and a weight decay of 0.001. To mitigate overfitting, we incorporate a dropout rate of 0.5 into the fully connected layer. During the fine-tuning process, we resort to a larger image size of 512 × 512 pixels, granting the model access to a richer set of details. Of those, 2629 images are used for training, 492 images are used for validating, and 164 images are used for testing. There is no overlap or mixing of data between the training and validation sets; all data in the validation set came from individuals who were entirely distinct from those in the training set. The training set was not exposed to the algorithm prior to the actual training process and stayed unprocessed and separated to ensure the model’s validity. In a bid to boost the model’s robustness, we engaged a suite of random augmentation techniques, such as horizontal/vertical flip, random cropping, rotation (up to 180 degrees), color adjustment (brightness, contrast, saturation, hue), grayscale application, and Gaussian blurring. Using the PyTorch framework, we applied on-the-fly data augmentation during each batch processing. This involves dynamically generating augmented images through transformations such as random rotations, flips, and color adjustments. These augmented images increase the diversity of the samples fed to the model during training. Training of the model is performed on an NVIDIA GeForce RTX 3080Ti GPU, which ensures swift training times because of its impressive memory and computational capacity.

### 2.4. Measurements Metrics

AUC Score: The area under the receiver operating characteristic curve (AUC) is a prevalent metric employed to evaluate binary classification models. It quantifies the model’s aptitude in differentiating between positive and negative examples across an array of thresholds. An ideal classifier commands an AUC of 1, whereas a random classifier holds an AUC of 0.5. The mathematical representation for AUC is as follows:(5)$$AUC=\frac{1}{2}\sum_{i=1}^{n-1}\left(x_{i+1}-x_{i}\right)\left(y_{i}+y_{i+1}\right)$$
where *n* is the total number of thresholds, and $x_{i}$ and $y_{i}$ correspond to the false positive rate and true positive rate, respectively, at the *i*th threshold.

Accuracy: Accuracy is a commonly deployed metric to assess classification models. It computes the ratio of accurately classified samples to the overall count of samples. The formula for accuracy is as follows:(6)$$Accuracy=\frac{TP+TN}{TP+TN+FP+FN}$$
where TP represents the count of true positives, TN stands for the count of true negatives, FP is the count of false positives, and FN refers to the count of false negatives.

F1 Score: The F1 score is an indicator of the equilibrium between precision and recall for binary classification models. It is the harmonic mean of precision and recall, and ranges from 0 (worst case) to 1 (best case). The formula for the F1 score is as follows:(7)$$F1=2\cdot \frac{precision\cdot recall}{precision+recall}$$
where precision is the ratio of true positives to the sum of true positives and false positives, and recall is the ratio of true positives to the sum of true positives and false negatives.

Precision and Recall: Precision and recall are two popular metrics employed for evaluating binary classification models. Precision computes the proportion of true positives out of the total count of predicted positives, and recall calculates the proportion of true positives out of the total count of actual positives. The formulas for precision and recall are given below:(8)$$Precision=\frac{TP}{TP+FP}$$
(9)$$Recall=\frac{TP}{TP+FN}$$
where TP denotes the count of true positives, FP is the count of false positives, and FN refers to the count of false negatives.

## 3. Results

### 3.1. Comparison with Other Methods

Table 1 illustrates the performance of the supervised baseline along with three self-supervised learning methods in detecting six prevalent diseases in UFI and CFI images. The self-supervised approaches exhibit superior mean AUC scores compared to the supervised baseline, underscoring the efficacy of self-supervised learning in medical image classification. Notably, our proposed technique yielded the highest AUC score of 86.96, outstripping the supervised baseline by 1.06%. This suggests that our technique is more proficient in identifying UFI diseases from fundus images. In addition, our method secured the best accuracy of 89.50%, F1 score of 62.51%, precision of 71.80%, and recall of 64.54% among all examined methods. These results extol the superior performance of our proposed approach compared to alternative self-supervised learning methods and the supervised baseline. The increased accuracy, F1 score, precision, and recall denote that our approach can discern diseases more precisely and dependably than other methods, a critical aspect for early disease diagnosis and effective treatment planning.

In Table 1, it is notable that the AUC score provides a measure of a model’s ability to distinguish between classes across various decision thresholds and is generally less affected by the class imbalance. In contrast, F1 score, precision, and recall are sensitive to the specific decision threshold applied and can vary based on the class distributions. Each instance in our dataset could belong to multiple classes, potentially leading to the variance in these metrics. The F1 score, precision, and recall metrics are influenced by the varying number of positive and negative samples for different classes. Even if a model has a high AUC or accuracy, these metrics can differ based on how the model performs on individual classes. Different models like no pre-train, supervised, SimCLR, Barlow Twins, and our proposed method may exhibit varying sensitivities to these factors. Understanding these nuances offers a more comprehensive view of the model’s overall performance and accounts for the observed disparities among the metrics.

Table 2 presents the mean AUC scores for different diseases achieved by various methods, including no pre-train, bi-lateral, multi-modality, and our proposed method. Our method consistently outperformed the other approaches across most diseases. Specifically, our method achieved the highest AUC scores for DM retinopathy (93.36), epiretinal membrane (84.73), and glaucoma suspect (97.29), demonstrating its effectiveness in accurately detecting these conditions. In addition, our method obtained competitive AUC scores for macular degeneration (87.39), retinal break (93.08), and retinal vein occlusion (69.49) compared to the other methods. These results highlight the accuracy of our proposed method in image classification, particularly for DM retinopathy and glaucoma suspects. The high AUC scores obtained by our method underscore its potential for improving the diagnosis and treatment of retinal diseases, contributing to enhanced patient care and outcomes.

The heatmaps generated by deep learning models can provide valuable insights into the areas of fundus images that are most indicative of a particular disease. In our study, we generated heatmaps for each of the six diseases in UFI images using our proposed contrastive learning method and compared them to those generated by a supervised learning method. Figure 6 demonstrates a compelling visual representation of the capabilities of our proposed model in comparison to conventional methods. Notably, the yellow boxes in the figure demarcate the areas correctly identified by our deep learning model as being afflicted by disease. These correctly identified regions reflect our model’s superior ability to interpret and understand the underlying features of fundus images, enabling it to pinpoint disease-afflicted areas with remarkable accuracy. On the contrary, the red boxes in the figure indicate the areas that were incorrectly identified by the traditional supervised learning model. This clear distinction between correctly and incorrectly identified regions reiterates the shortcomings of the supervised learning models, particularly in comparison to the advanced capabilities of our proposed model. In our study, we utilized distinct methods for visualization and interpretability of the learned representations, specifically employing Grad-CAM for the models utilizing InfoNCE loss and Score-CAM for the Barlow Twins model, since Barlow Twins uses a different loss function than the other methods (Barlow Twins loss). Our deep learning model exhibits a clear capacity to identify all disease-afflicted areas of the fundus image, a feat which conventional supervised learning models fell short of achieving. A detailed evaluation of our results reveals that self-supervised learning methods, and, more specifically, our unique contrastive learning method, generate heatmaps that exhibit an enhanced degree of accuracy, zeroing in on the disease-related regions with unparalleled focus when compared to the heatmaps generated by supervised learning methods.

The performance of different models for classifying diseases was evaluated using receiver operating characteristic (ROC) curves, as shown in Figure 7. The ROC curves provide an effective way to visualize the trade-offs between the true positive rate (sensitivity) and the false positive rate (1-specificity) across different threshold settings. Each curve represents a different model’s ability to discriminate between positive and negative instances for each disease class. Models that closely follow the upper left border of the ROC space indicate a better performance, whereas models that lie closer to the diagonal line reflect less discriminative power. The area under the curve (AUC) values further quantify the overall ability of each model to distinguish between the classes, with higher AUC values suggesting better model performance. In addition to the ROC curves, the performance of our proposed model was also evaluated using confusion matrices for every disease class, shown in Figure 8. A confusion matrix offers a more granular view of the model’s performance, breaking it down into true positives, true negatives, false positives, and false negatives. This allows for the calculation of various metrics, such as sensitivity, specificity, and F1-score. All of the models exhibit high true positive rates across multiple classes, indicating good sensitivity in detecting diseases. At the same time, the true negative rates are substantial, reflecting that the models are adept at correctly identifying non-disease instances. It is worth noting that the total number of instances in each confusion matrix remains constant across different classes, as each instance may belong to multiple classes in our multi-label, multi-class setting.

Figure 9 presents the training and validation loss curves for five different models: random initialization, supervised learning, SimCLR, Barlow Twins, and our proposed method. Each curve represents the model’s loss over a series of training epochs, offering insight into the model’s learning efficiency and its ability to generalize. Most notably, the curve for our proposed method consistently exhibits the lowest loss values across the epochs and demonstrates a continuous downward trend, indicating optimal learning and generalization capabilities. This performance underscores the efficiency of our method in minimizing the loss function compared to other established techniques. In particular, the declining nature of the loss curve for our proposed method suggests that the model is effectively learning the underlying patterns in the data without signs of overfitting. This outcome is further supported by our use of weight decay as a regularization technique and periodic validation every iteration.

### 3.2. Ablation Study

Table 3 shows a comparative analysis of the performance of different self-supervised learning methods in disease detection. We utilized several metrics for this comparison, namely the area under the curve (AUC) score, accuracy, F1 score, precision, and recall. Starting with the SimCLR method, it yielded an AUC score of 85.09, an accuracy of 88.25%, an F1 score of 57.42, precision of 67.70, and a recall of 64.18. Bi-lateral learning showed a slightly better AUC score of 85.47, although its accuracy was slightly lower at 88.13%. The F1 score for bi-lateral was 57.16, with a precision of 63.18 and recall of 63.93. The multi-modal learning method presented a more significant improvement, with an AUC score of 85.90, accuracy of 89.13%, an F1 score of 61.54, precision of 67.99, and a recall of 64.19. However, our proposed method clearly outperformed the others. It achieved the highest AUC score of 86.96, an accuracy of 89.50%, F1 score of 62.51, precision of 71.80, and a recall of 64.54. These results provide empirical support to our claim that our proposed self-supervised learning model offers a substantial improvement over existing methods across multiple performance metrics in classification.

To examine the impact of various self-supervised method combinations on image classification efficiency, we conducted an ablation study, integrating distinct self-supervised methods. Each combination is trained with the same settings and environment as our proposed method, but with different datasets and loss. Specifically, method 1 + 2 uses $L_{pair-instance}+L_{Bi-lateral}$, method 2 + 3 uses $L_{Bi-lateral}+L_{multi-modality}$, and method 1 + 3 uses $L_{pair-instance}+L_{multi-modality}$. Table 4 encapsulates the performance of these combinations in terms of mean AUC, accuracy, F1 score, precision, and recall. Our study demonstrated that the triumvirate of pair-instance, bi-lateral, and multi-modality procured the superior mean AUC score of 86.96. This performance slightly edged out the amalgamation of SimCLR and bi-lateral (86.75) as well as the bi-lateral and multi-modality ensemble (86.51). However, the fusion of bi-lateral and multi-modality secured the leading F1 score of 63.06, demonstrating an optimal balance between precision and recall. Furthermore, the pair-instance and bi-lateral combination achieved the highest recall score of 67.88, signifying an enhanced capacity to detect positive instances. The result of this exploration suggests that blending multiple self-supervised methods can bolster the performance of the model. Furthermore, these combinations could establish diverse trade-offs among different evaluation metrics. This informs us that choosing the right blend of methods may be application-specific, whereby the optimal combination may differ based on the diagnostic criteria of the disease or the medical scenario at hand. For example, in cases where a high recall is imperative to avoid missing any potential positive cases (in conditions with severe consequences or when early diagnosis significantly improves prognosis), the pair-instance with bi-lateral combination may be most suitable. Conversely, when it is crucial to have a balanced model in terms of both recall and precision (to avoid an excessive number of false alarms or when resources for further testing are limited), the bi-lateral and multi-modality pair could be the preferred choice. Such insights, derived from our ablation study, are instrumental in tailoring self-supervised learning models to particular disease diagnosis tasks, thereby advancing precision medicine and contributing to better patient outcomes.

### 3.3. Contribution of Each Pairing Method on the Improvement of the Performance

To quantify the individual improvement brought by each method, we subtract the baseline performance metrics (termed as “no pre-train”) from the metrics achieved when the respective method was applied. Formally, for a given performance metric *M* and a pre-training method $M_{i}$, the individual improvement $I_{i}$ is calculated as:(10)$$I_{i}=M_{i}-M_{\mathrm{No}\;\mathrm{Pre}-\mathrm{train}}$$
This calculates the net increase in performance solely due to the implementation of a particular pre-training strategy. Subsequently, we determine the incremental benefit provided by a method when it is amalgamated with another. This additional improvement $A_{ij}$ for a pair of methods $M_{i}$ and $M_{j}$ is computed as:(11)$$A_{ij}=M_{i+j}-M_{j}$$
This measures the extra performance enhancement due to the synergistic interaction of the two methods. For instance, when calculating the improvement introduced by the pair-instance method in conjunction with the bi-lateral method, the additional improvement is obtained by subtracting the performance of the bi-lateral method alone from the performance delivered by the combination of the pair-instance and bi-lateral methods. This calculation manifests the additional contribution the pair-instance method proffers when paired with the bi-lateral method. We repeat this process for all pairs and average the additional improvements. The total contribution $C_{i}$ of a method is then computed as the sum of its individual improvement and the mean of its additional improvements when combined with the other methods:(12)$$C_{i}=I_{i}+\frac{1}{n}\sum_{j}A_{ij}$$
This formula ensures that we capture the total contribution of each method, considering both its standalone impact and the augmentations it imparts when coupled with others. Figure 10 shows a detailed comparative analysis of the performance improvements delivered by our three individual self-supervised learning methods—pair-instance, bi-lateral, and multi-modality—against a baseline model with no pre-training. The performance improvements were calculated based on five evaluation metrics: AUC score, accuracy, F1 score, precision, and recall. To ascertain the performance improvement for each method, we subtracted the metric score of the baseline model from the score achieved by each method. The result is a measure of the performance gain attributable to the incorporation of each self-supervised learning method. The bar graph in the figure visually showcases the performance enhancements for each method. As per the numerical values indicated, the multi-modality method registers the highest improvement across all performance metrics, implying that this method is the most beneficial of the three. It is closely followed by the bi-lateral method, which demonstrates substantial improvements as well, albeit slightly less than the multi-modality approach. The pair-instance method also outperforms the baseline, but its gains are comparatively modest. These results reinforce the merit of our proposed self-supervised learning strategies in enhancing the accuracy of deep learning models, surpassing the non-pretrained model in all respects. The bar graph in the figure visually showcases the performance enhancements for each method. Significantly, the multi-modality method registers the highest improvement across all performance metrics. This dominant contribution underscores the importance of correlating multiple modalities within medical images, emphasizing that the interplay between different image forms is vital in capturing intricate and critical features that enhance diagnostic accuracy. The bi-lateral method, though marginally surpassed by the multi-modality approach, also demonstrates substantial improvements. It leverages the left and right perspectives of anatomical structures within medical imaging, allowing the model to grasp nuanced differences and gain a holistic understanding. Such bi-lateral insights into the data structure enable the model to recognize intricate patterns that would otherwise be elusive, thereby significantly boosting its performance.

## 4. Discussion and Conclusions

In conclusion, we introduced a contrastive learning approach for pre-training deep neural networks in the context of retinal disease diagnosis. Our proposed approach leveraged the power of three different types of image pairs, including pair-instance, bi-lateral, and multi-modality, to create a diverse and comprehensive set of positive and negative pairs for contrastive learning. Our experimental results demonstrated that our method surpassed the performance of the supervised learning baseline, achieving state-of-the-art results across four major retinal disease diagnosis tasks. These findings underscore the effectiveness of self-supervised learning in the field of retinal imaging, highlighting its potential for advancing early diagnosis and treatment strategies.

The strength of our findings lies in their potential to revolutionize early diagnosis and treatment planning for retinal diseases. The application of self-supervised learning, as demonstrated in our study, brings a new dimension to the field by addressing challenges such as limited labeled data and manual annotation effort. This has profound implications for not only improving diagnostic accuracy but also for reducing the cost and time involved in retinal disease management. In medical fields where early diagnosis can dramatically impact patient outcomes, any advancement in accuracy is particularly crucial. Our method’s enhanced performance could potentially lead to more effective treatments and interventions, ultimately improving patient prognosis and quality of life.

Furthermore, our work underscores the potential of self-supervised learning to revolutionize medical imaging, offering a flexible and data-efficient alternative to traditional supervised approaches. We acknowledge that ongoing research is essential to explore the method’s generalization potential across diverse clinical scenarios and diverse datasets. Our study also opens avenues for incorporating additional modalities and clinical data, deepening the synergy between machine learning and medical expertise. Although our method shows promise, it is not without limitations.

The study was focused on a specific subset of retinal diseases, and the algorithms have yet to be validated on a diverse demographic or against rare retinal conditions. Another limitation of this study is the relatively small size of the dataset used for training and validating the models. The paucity of data may not capture the full complexity and variability associated with medical fundus images and the diseases they are meant to help diagnose. This limitation may affect the generalizability of our model to a broader, more diverse population. Future research could benefit from utilizing larger and more diverse datasets to validate the effectiveness of the proposed algorithm and to potentially improve its diagnostic capabilities. Furthermore, a comprehensive hyperparameter optimization was not conducted. Future work will focus on a more rigorous optimization of the model’s hyperparameters, employing techniques such as grid search or Bayesian optimization.

Moving forward, our future work will explore the impact of incorporating different modalities and data sources in our proposed method. For instance, alongside fundus images, we aim to incorporate additional imaging modalities such as optical coherence tomography (OCT) and angiography to create multi-modal image pairs [27,28]. Furthermore, we are interested in investigating the integration of other modalities, such as clinical data and genetic information, in conjunction with retinal images to enhance the model’s accuracy [29]. Additionally, we plan to assess the generalizability of our method to other medical imaging domains beyond retinal disease diagnosis [30]. As deep learning models become more complex, there is also a need to develop efficient and scalable self-supervised learning methods capable of handling large-scale medical image datasets [31]. We anticipate that our work will inspire future research endeavors in this direction, ultimately contributing to improved disease diagnosis and treatment outcomes.

## Figures and Tables

**Figure 1 bioengineering-10-01089-f001:**
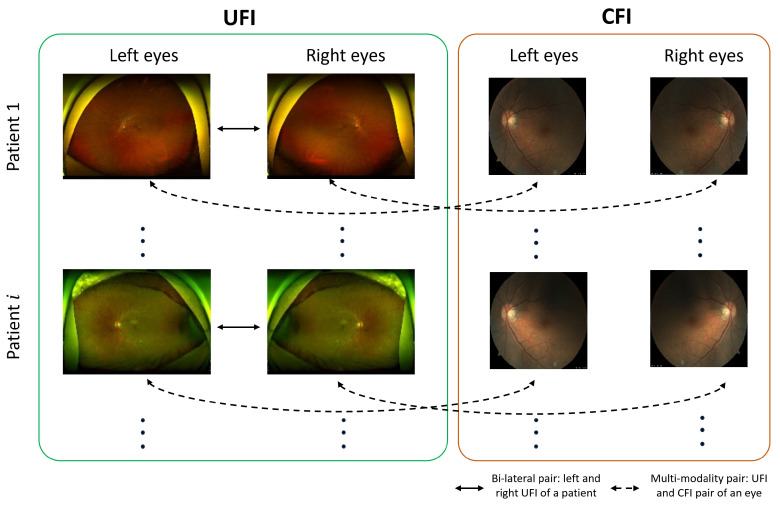
Implicit data correlation between fundus images. Bi-lateral and multi-modality pairs are positive samples, and images from different patients are considered as negative samples.

**Figure 2 bioengineering-10-01089-f002:**
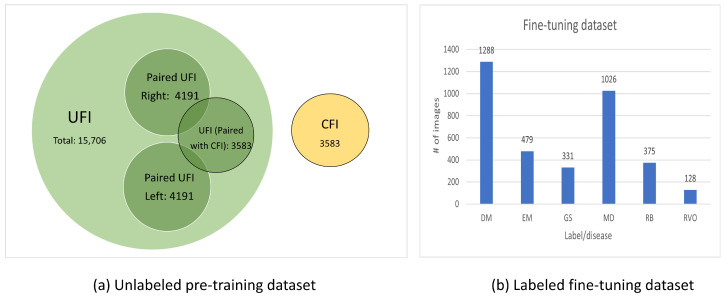
Summarization the of pre-training and fine-tuning dataset. DM: DM retinopathy, EM: epiretinal membrane, GS: glaucoma suspect, MD: macular degeneration, RB: retinal break, RVO: retinal vein occlusion.

**Figure 3 bioengineering-10-01089-f003:**
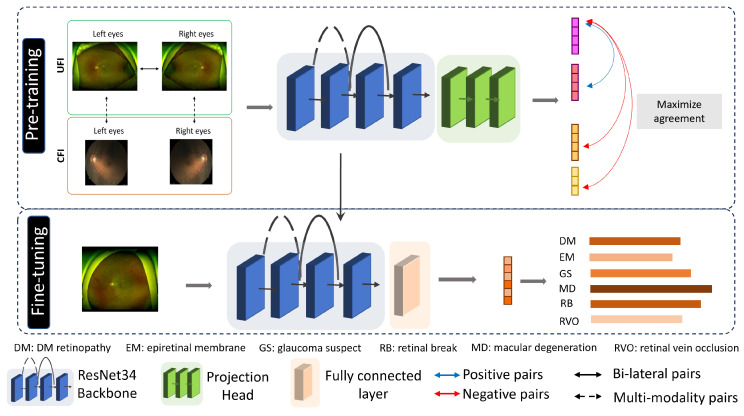
The general process of classification process on fundus image using contrastive pre-training. Positive and negative pairs are used to pre-train the model backbone, then the backbone is used for fine-tuning with labeled data.

**Figure 4 bioengineering-10-01089-f004:**
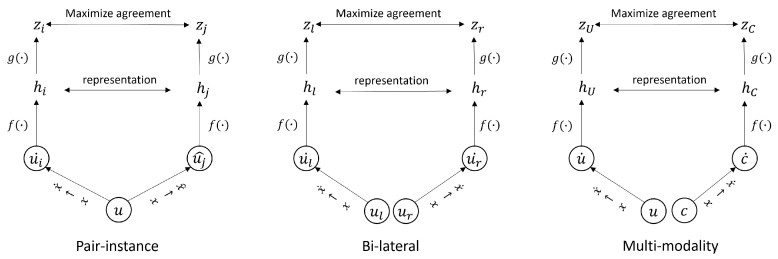
The framework of contrastive learning for pair-instance, bi-lateral, and multi-modality pre-training. The images are augmented to form positive views (for pair-instance) and new augmented view (for bi-lateral and multi-modality). The encoder *f* and projection head *g* try to maximize the agreement using contrastive loss); $x\to \dot{x}/\hat{x}$ denotes the data augmentation step.

**Figure 5 bioengineering-10-01089-f005:**
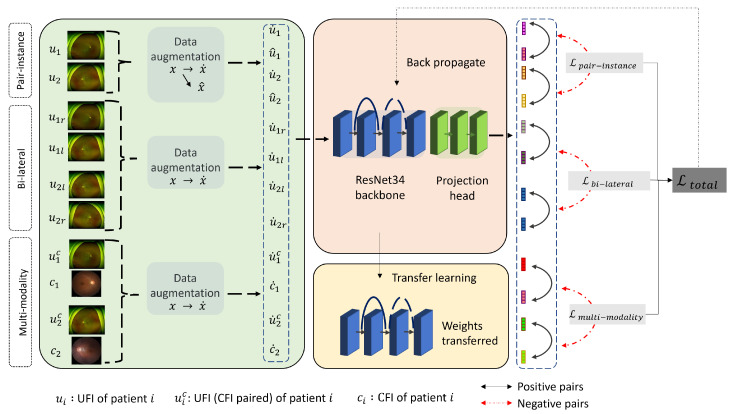
Overview of our proposed multi-modality contrastive learning method. The model is pre-trained on a combination of UFI and CFI using pair-instance, bi-lateral, and multi-modality techniques.

**Figure 6 bioengineering-10-01089-f006:**
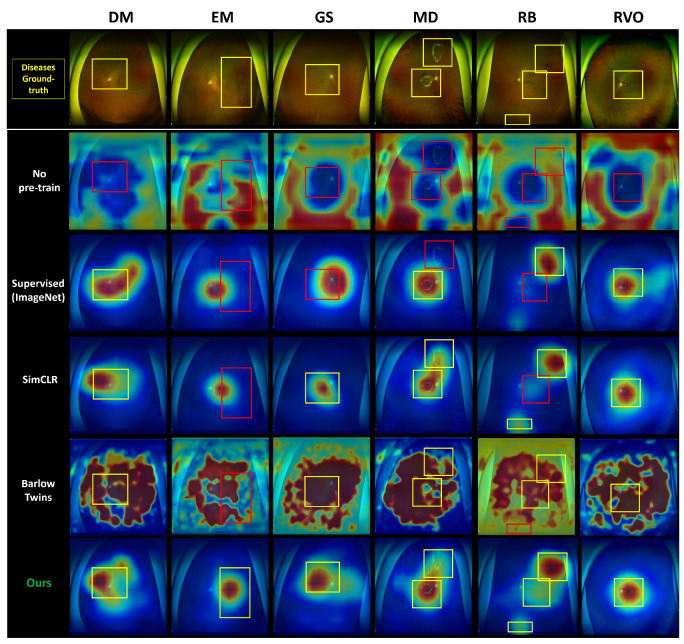
Side-by-side comparison of the models’ heatmaps, including traditional supervised learning and our proposed contrastive learning model. Yellow boxes represent accurately identified disease-afflicted areas, demonstrating the precision of our model. Conversely, red boxes indicate the areas incorrectly identified by the traditional model.

**Figure 7 bioengineering-10-01089-f007:**
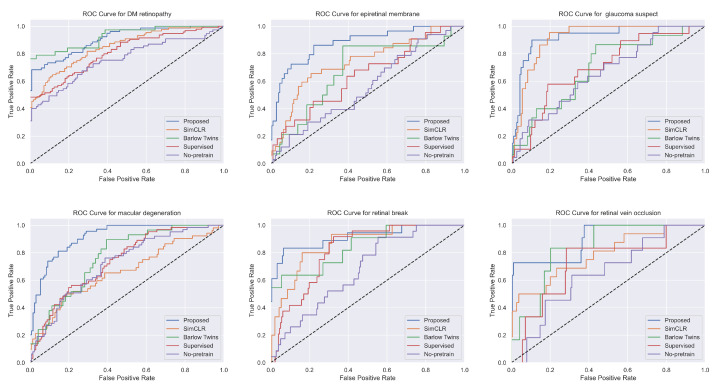
ROC curves of different models for each disease. The dashed line in the middle indicates a random classifier. The curve that is further to the left corner indicates a better model.

**Figure 8 bioengineering-10-01089-f008:**
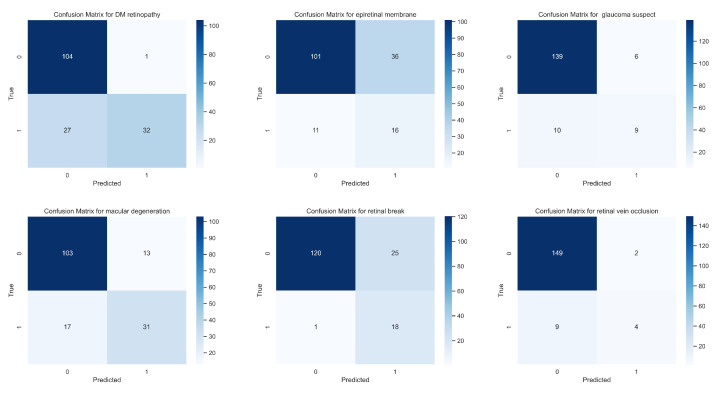
The confusion matrix for each disease of our proposed model.

**Figure 9 bioengineering-10-01089-f009:**
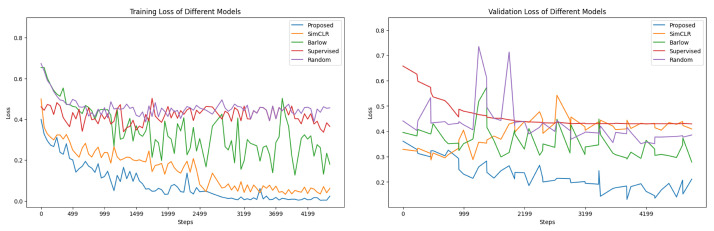
Training and validation loss of different models. The figure shows the convergence of training and validation loss over epochs, indicating a low likelihood of overfitting.

**Figure 10 bioengineering-10-01089-f010:**
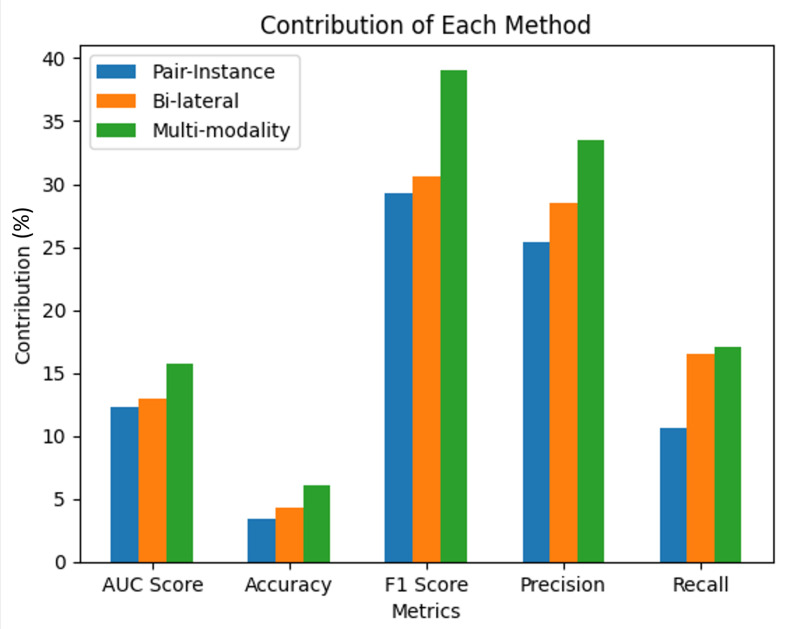
Comparative performance improvement of the three individual self-supervised learning methods—pair-instance, bi-lateral, and multi-modality—over the baseline model (no pre-train). Each bar represents the increase in the respective performance metric (AUC score, accuracy, F1 score, precision, recall) attributable to the implementation of the corresponding method.

**Table 1 bioengineering-10-01089-t001:** Performance of different approaches on image classification, including random initiation weights, supervised learning, simCLR, and our proposed method.

Methods	AUC Score	Accuracy	F1 Score	Precision	Recall
No pre-train	72.95	84.21	28.29	39.47	48.29
Supervised (ImageNet)	84.57	87.67	57.11	67.03	61.22
SimCLR [18]	85.09	88.25	57.42	67.70	64.18
Barlow Twins [26]	85.47	89.46	40.14	55.93	53.69
Ours	**86.96**	**89.50**	**62.51**	**71.80**	**64.54**

Texts in bold denote the best performance.

**Table 2 bioengineering-10-01089-t002:** Mean AUC score of each disease.

Methods	DM	EM	GS	MD	RB	RVO
No pre-train	74.39	54.24	84.48	72.47	68.42	64.75
Supervised (ImageNet)	91.16	57.56	83.31	84.62	92.66	**90.05**
SimCLR [18]	90.31	79.48	86.19	84.63	92.77	66.68
Barlow Twins [26]	**95.65**	64.52	83.17	86.79	85.45	83.93
Ours	93.36	**84.73**	**97.29**	**87.39**	**93.08**	69.49

DM: DM retinopathy, EM: epiretinal membrane, GS: glaucoma suspect, MD: macular degeneration, RB: retinal break, RVO: retinal vein occlusion. Texts in bold denote the best performance.

**Table 3 bioengineering-10-01089-t003:** Performance of different self-supervised learning methods on image classification.

Method	AUC Score	Accuracy	F1 Score	Precision	Recall
Pair-instance	85.09	88.25	57.42	67.70	64.18
Bi-lateral	85.47	88.13	57.16	63.18	63.93
Multi-modality	85.90	89.13	61.54	67.99	64.19
Ours	**86.96**	**89.50**	**62.51**	**71.80**	**64.54**

Texts in bold denote the best performance.

**Table 4 bioengineering-10-01089-t004:** Performance of supervised baseline and three self-supervised learning methods on image classification.

Pair-Instance	Bi-Lateral	Multi-Modality	AUC Score	Accuracy	F1 Score	Precision	Recall
✓	✓		86.75	88.92	59.55	68.77	62.08
✓		✓	86.55	88.71	61.19	66.46	61.99
	✓	✓	86.51	88.83	**63.06**	70.53	**64.94**
✓	✓	✓	**86.96**	**89.50**	62.51	**71.80**	64.54

Texts in bold denote the best performance.

## Data Availability

Data available on request (data availability is decided by the IRB of Kangbuk Samsung Hospital on each request).

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
