# Peer review of "Self-FI: Self-Supervised Learning for Disease Diagnosis in Fundus Images"

_bioengineering, 2023, doi:10.3390/bioengineering10091089_

Round 1

Reviewer 1 Report

The paper submitted by the authors is quite exciting and provides relevant information about the utility of unlabeled images and self-supervised learning strategies in this field. However, I do not feel comfortable about the data collected by the authors and how they received approval from the ethical committee. Therefore, and only for this reason, I recommend the rejection of the paper.

Minor concerns 

Please in the abstract section explains the AUC abbreviation meaning.
Lines 29-30, please add a reference.

Please explain better legend of Figure 1.

Line 48, what do you mean here with natural images? I suppose you mean ocular, right? Please clarify.

Line 71, which one?

Line 114, I do not agree, I cannot see how the ethical committee waived it. Then, you must present the arguments and the permission provided by the committee.

Line 137, correct.

Line 140, in my opinion, this sentence doesn’t reflect that statement about Figure 2, better line 145.

In Figure 3, add the information about the different abbreviations presented on it.

Lines 233-238, please check because this paragraph is repeated.

Line 301, is not 164 a low number For testing?

The discussion section is relatively poor taking into consideration your relevant results. Thus, I recommend enriching it.

Reviewer 2 Report

The manuscript proposes a so-called Self-FI neural network to classify fundus images into several classes. Bi-lateral and multi-modality pre-training techniques are used to train a ResNet34 backbone neural network before fine-tuning. The fundus image is classified into one of the six diseases. 

The idea of the contrastive learning in this study seems interesting but it is a puzzle to the reviewer why the bi-lateral and multi-modality pre-training can help improve the quality of disease classification. Although the authors try to explain that the structure of the feature space may be better learned with the contrastive learning techniques, why are both necessarily related to the disease classification? They may not share the same features at all. The ablation experiment results also indicate that both techniques do not contribute too much in performance improvement, even if the difference in the assessment metrics are statistically significant. 

The evidence of the sufficient training of the neural network is lacking. How does not loss function converge during the training process? Why likely does the network overfit the training dataset? Are there enough results to show readers that the hyperparameters are optimized in any sense?

If even the neural network is well-trained, the significance of the improvement in the performance is suspicious. How is the AUC calculated for multi-class classification? In Table 1, both the AUC and Accuracy seem different significantly but the F1 score and other metrics change a lot. It is contra-intuitive and an explanation may be necessary to help readers understand it. 

Another confusion is from Fig. 5, in which the authors claim that the diseases can detected with the proposed neural network. The reviewer believes the difference between classification and detection is clear and a classifier cannot do disease detection for sure. 

In Fig. 6, what is the unit of the y axis?

Discussion of the study is too short.

There are many fundus image classification techniques out there in the literature, why only two, namely SimCLR and Barlow Twins, are used in the study for comparison? 

In addition, the reviewer also believes there is a clear cut between image classification and disease diagnosis. Please do not confuse these two different concepts in the manuscript.

English is OK except for some grammatical errors here and there.

Reviewer 3 Report

A lot of work here, but the paper has major problems that would require, at least, a major rewrite.

Major

1. The methodology is poorly described, I doubt that any reader would be able to apply the method, let alone replicate the work. Equations and algorithms too briefly described to be understandable. There is no point in publishing a paper that nobody can understand, about a model that lives only on the hard drive of the author.

2. Validity of the work is not justified, and I strongly suspect that the apparently good performance is the result of overfitting. The "n" seems far too small to allow a 6 way classifier to be trained.  The dataset contains *no* normals, a significant limitation.

lines 300-306 (augmentation of images) - how many augmented images were created?  Please state explicitly that there was no mixing of data between the training and validation sets - none of the data in the validation set came from individuals in the training set.

Please state explicitly that the training set was not exposed to the algorithm in any way (a necessary requirement for validity). Those data had to be "virgin", unprocessed in any way otherwise significant leakage will occur.

2. Presentation of results. The AUC is inadequate., The accuracy in estimating the AUC depends on how many points on the ROC are examined, which is not stated (i.e. what is n in Eq. 4)? If, as I suspect, the AUC was calculated based on only one or two thesholds, the small differences between AUCs that are the chief finding of the study are totally insignificant.

If the authors quote AUCs please show the corresponding ROCs to let readers judge how reliable this metric is likely to be in distinguishing the performance of the different methods.

Table 1 - I simply do not believe that an algorithm giving an AUC of 89.96 is significantly better than one with an AUC of 85.47.

3. Please show a confusion matrix for the 6 conditions showing the numbers of patients that had been correctly and incorrectly diagnosed., That is more relevant to the stated goal of the work, diagnosing disease.  

Minor:

did the authors use "canned" software, such as python packages? If so please cite them.

Round 2

Reviewer 1 Report

Dear Authors,

The comments have been properly addressed. However, and before acceptance, the authors should upload the file received by the Ethical Committee

Author Response

Please kindly find the attachment for the document from the ethical committee. Thank you very much for your time.

Reviewer 2 Report

Thanks for the efforts. I have no further comments.

Author Response

Thank you very much for your time.

Reviewer 3 Report

Authors have satisfactorily responded to comments in first round of review, paper is significantly improved, does not need further revision in my view. 

Author Response

Thank you very much for your time.